# The effect of malnutrition on adult Covid-19 patient's ICU admission and mortality in Covid-19 isolation and treatment centers in Ethiopia: A prospective cohort study

**Lencho Mekonnen Jima**[1], **Gudina Egeta Atomsa**[1], **Johane P. Allard**[2], **Yakob Desalegn Nigatu**[1] *

1 Department of Nutrition and Dietetics, School of Public Health, Addis Ababa University, Addis Ababa, Ethiopia, 2 Department of Medicine, University Health Network, University of Toronto, Toronto, Canada

* yakobdesalegne@gmail.com

**Data Availability Statement:** All relevant data are within the manuscript and its Supporting Information files.

## Abstract

### Background

A new coronavirus was first identified in Wuhan, China in December 2019. Since the times of the 1918 influenza pandemic, malnutrition has been known as a risk factor for severity and mortality from viral pneumonia. Similarly, the recently identified SARS-Cov2 infection (COVID-19) and related pneumonia may be closely linked to malnutrition. Therefore, this study will contribute to new knowledge and awareness of the recording and evaluation of each COVID-19 patient's nutritional status by assessing the effect of malnutrition on ICU admission and death of COVID-19 patients in developing countries.

### Method

We conducted a prospective cohort study in adult COVID-19 patients admitted to selected COVID-19 Isolation and Treatment Centers, Addis Ababa, Ethiopia. Baseline data of the patients were collected using interviewer-administered structured questionnaire and data on the adverse outcomes of follow up were extracted from follow up chart. The main clinical outcomes (ICU admission and death) were captured every week of follow up. We ran a multivariate Cox's regression analysis to determine the relationship between malnutrition at admission and its effect on ICU admission and death.

### Results

A total of 581 COVID-19 patients were enrolled. From the total of recruited patients, 346 (59.6%) were males and 235 (40.4%) were females. The mean age of the respondents was 55 years (16.45) years. The Cox proportional hazard model controlled for sex, age group, number of co-morbidities, and number of medications found that malnutrition at admission was associated with ICU admission and death. When compared to well-nourished patients, the rate of ICU admission was significantly associated and found to be higher among underweight [(adjusted hazard ratio (AHR) = 10.02, 95% CI: (8.64–12.10)] and overweight [(AHR

**Funding:** This study was supported by Addis Ababa University. The funding body had no role in designing the study, in the collection, analysis and interpretation of data, decision to publish, in the writing of this manuscript and in the decision to submit for publication.

**Competing interests:** The authors have declared that no competing interests exist.

= 7.7, 95% CI: (6.41–9.62)] patients. The rate of survival probability was significantly associated and was found to be better among well-nourished patients (AHR = 0.06, 95% CI : (0.01–0.44) when compared with malnourished COVID-19 patients.

## Conclusion

Malnutrition at the time of admission was shown to increase the risk of ICU admission and mortality among COVID-19 patients. Therefore, it is vital to evaluate patients' nutritional condition early in their admission and provide timely intervention to minimize the effects on patients and the healthcare system.

## Introduction

A new coronavirus was first identified in China in December 2019, causing an outbreak of pneumonia (COVID-19). Although most patients present with mild symptoms and quickly recover, the disease can cause severe respiratory disease, multiple organ dysfunction, and even death [1]. Since the times of the 1918 influenza pandemic [2], malnutrition has been known as a risk factor for severity and mortality from viral pneumonia. Similarly, the recently identified SARS-Cov2 infection and related pneumonia (COVID-19) may be closely linked to malnutrition [3, 4]. Indeed, a variety of characteristics found in COVID-19 patients are likely to result in body weight loss and malnutrition [5]. COVID-19 is a condition marked by an inflammatory syndrome, contributing to decreased food consumption and enhanced muscle catabolism, which is why patients with COVID-19 are at an elevated risk of malnutrition, making avoidance of malnutrition and dietary control important aspects of treatment [6].

In Ethiopia, having different nutritional problems such as underweight, micronutrient deficiency, and being overweight can all have an impact on COVID-19 patient adverse outcomes. Poor nutritional status with an increment of the prevalence and severity of malnutrition among COVID-19 patients can lead to a higher rate of intensive care unit (ICU) admission and longer stays in intensive care [7, 8]. Similarly, an overweight and obese COVID-19 patient raises the risk of hospitalization, ICU admission, intubation, and death [9–13].

As the result of literature and prior research indicates, being a malnourished COVID-19 patient can lead to a higher rate of ICU admission and increased death. Given the limited evidence in Africa, this research assesses the effect of malnutrition on ICU Admission and death of COVID-19 patients to fill the gap in knowledge on the documentation and assessment of the nutritional status of each COVID-19 patient and the lack of adequate attention to the nutritional intervention of COVID-19 patients.

## Materials and methods

### Study setting

The study was conducted in the COVID-19 Isolation and Treatment Center of St Paul's Hospital Millennium Medical College (SPHMMC), Eka Kotobe General Hospital (EKGH) and Millennium COVID-19 care center (MCCC). St. Paul Millennium Medical College Hospital is located in the capital city of Ethiopia, Addis Ababa. It is the second largest hospital in Ethiopia. Eka Kotobe General Hospital was the first hospital assigned to treat positive cases of COVID-19 in Ethiopia. It has the potential to admit 600 cases. It is located in Addis Ababa. Whereas

Millennium COVID-19 care centers are temporary establishments prepared to handle an influx of COVID-19 cases.

## Study design

We conducted a facility-based prospective cohort study from December 2021 to February 2022 in selected COVID-19 Isolation and Treatment Centers.

## Eligibility criteria

All individuals included in this study tested positive for COVID-19 through confirmed rRT-PCR (real-time reverse transcriptase Polymerase Chain Reaction) testing and were aged 18 years or older. Participation required the provision of consent without further symptomatic evaluation for COVID-19. Exclusions from the study comprised critically ill patients with bilateral upper extremity amputations, as this condition posed challenges in assessing nutritional status upon admission

## Sample size calculation

Sample size was calculated to determine the magnitude of effect between malnutrition at admission and adverse outcomes of COVID-19.We used a double population proportion formula and the following assumptions to calculate sample size to determine the relationship between malnutrition at admission and adverse outcomes; P1 (Proportion of ICU admission among malnourished patients) = 9.5%, P2 (Proportion of ICU admission among well-nourished patients) = 9.5% [14]. Also P1 and P2 assumed for death P1 (Proportion of death among malnourished patients) = 1.5%, P2 (Proportion of death among well-nourished patients) = 1.5% [15], $Z \alpha/2$ = 1.96 at a 95% level of confidence, 5% margin of error. And r = 1 (the proportion of ICU admission and death among malnourished and well-nourished patients was taken as equal, or 1:1) and 10% non-response rate. Based on this assumption, a total of 581 COVID-19 patients were required in the study based on the highest sample size obtained from ICU admission prevalence.

## Sampling procedures

The study was conducted at EKGH, MCCC, and SPHMMC, COVID-19 Isolation and Treatment Centers using a purposive sampling method. These selected centers are mainly working on COVID-19 patients with well-equipped ICU setup and larger admission capacity. Sample was collected proportionally from each center based on their admission capacity until the minimum sample size of the study fulfilled. Consecutive sampling method was used to enroll study participants by keeping inclusion and exclusion criteria consecutively until expected number of sample size was reached.

## Data collection

The base line data were collected using interviewer administered questionnaire by a nurse who is working in triage and also objective weight and height measurement was taken place using electronic portable weight scale (seca scale) and stadio-meter for height. The principal investigator provides training, technical assistance, and measurement equipment to data collectors in each center. MUAC was measured on the non-dominant arm using non-stretchable plastic tape at the midpoint between the olecranon and the acromial process after the arm is flexed to 90 degrees at the elbow. To improve intra-observer reliability, data collectors were trained in relevant measuring techniques by the principal investigator until an acceptable level of

technical error of measurement (TEM) was obtained. The technical error of measurement and coefficient of variation of weight, height, and girth of limbs (MUAC) for both inter-examiner and intra-examiner measurements were all within acceptable limits. This means, as indicated in the literature, acceptable TEM values should be of the order of 0.1 kg for weight, 3 mm for height, and 2 mm for girth of limbs [16]. Before starting the measurement, the measurement tools were calibrated. Also, body weight was measured to the nearest 0.1 kg with light clothing and no shoes, using an electronic portable scale (seca scale). To ensure measurement accuracy, the scale was checked for a zero reading before each weighing. Both MUAC and height were measured to the nearest 0.1 cm with a flexible non-stretch tape and a studio meter, respectively. Those used to categorize nutritional status as follows. Undernourished: COVID-19 case with BMI $<18.5$ kg/m$^2$ and for those critically ill patients who are not capable of measuring BMI the MUAC measurement of $<23.7$cm were considered as underweight [17]. Overnourished: COVID-19 case with BMI $>25$ kg/m$^2$ and for those critically ill patients who are not capable of measuring BMI the MUAC measurement of $\geq28.1$cm were considered as overweight [17]. MUAC is also recommended for patients with ascites or edema in legs or trunk to gauge dry weight and BMI [18]. Other studies confirm that MUAC correlated well with BMI and could be used to identify patients as underweight and overweight [19–22].

In addition, a periodic supervision was held by nutritionist who is working in hospital to gate the adverse outcomes of the patient every week during follow up. ICU admission and death were censored at 2 months, hence COVID-19 patients who were discharged earlier than follow-up time without adverse outcome or stayed in the hospital longer than 2 months were treated as censored observations.

## Statistical analysis

Before entering the data, the questionnaire was reviewed for completeness. Data entry was carried out on a kobo toolbox. Then exported into Excel 2007 for cleaning and coding [see S1 File]. Finally, structured data was analyzed using the Social Sciences Statistical Package (SSPS) version 23. The patient cohort characteristics for continuous data were measured in terms of central tendency (mean or median) and dispersion (standard deviation or range). A frequency distribution was used for categorical data. A chi-square test was used to show variables having a statistically significant association with the nutritional status of the patients. A chi-square test of association was done between nutritional status and other variables. As a result, those having a p-value less than 0.05 were included in the analysis of Cox regression.

Descriptive analysis of survival data was presented graphically using the Kaplan-Meier (KM) estimator.

In accordance with the proportional hazard's assumption, our analysis reveals that the hazard ratios for the examined variables remain constant over time. This finding is supported by the examination of Schoenfeld residuals and log-log plots, which indicate parallelism between survival curves. A Log-Rank test was used to compare the survival experiences of different categories of covariates. Multivariable Cox-regression analysis was used to control for possible confounding factors and identify risk factors for COVID-19 adverse outcomes. The adjusted hazard ratio (AHR) along with its 95% confidence interval was estimated to measure the strength of association. The level of statistical significance was set at a p-value less than 0.05. While checking multicollinearity, the result of variance inflation factor (VIF) value was near to 1, which indicates that there was no correlation between each independent variable.

## Ethical consideration

The study received ethical approval from the Research Ethical Committee of the School of Public Health, Addis Ababa University. Also, from each COVID-19 care center's institutional review board (IRB), an ethical clearance was obtained before data collection took place. During the data collection, written informed consent was obtained from each patient after explaining the objectives of the study. To maintain anonymity, identifiers like names were not included in the questionnaire. All measures to maintain human rights, including informed consent, the right to participate in the study, the right to privacy and confidentiality, and the right to prevention of any type of harm, were taken into consideration.

# Result

## Socio-demographics characteristics

A total of 581 patients were studied. Of these, nutritional status was assessed for 22% of the patients using MUAC, and for the remaining 78% using BMI. There was no case in children and adolescence while 174(29.9%) of cases were 65 years of age or older. The mean (±SD) age was 55 (±16.45) years old with 346(59.6%) being male and 235(40.4%) female, 527 (90.7%) were urban residents and 495 (85.2%) were married. The mean (±SD) monthly incomes of the respondents were 4531(±3642) birr and range was between 0–20,000 birr [see S1 Table].

## Admission-related clinical characteristics of the patients

Table 1 shows admission-related clinical characteristics of the patients by malnutrition status. Among the participants, 289 individuals exhibited malnutrition, encompassing both over nutrition (204 cases) and under nutrition (85 cases), while 292 participants were classified as well-nourished. There were statistically significant association seen between nutritional status younger age, female sex, income, co morbidity and higher number of medication and. In the study population, 45% of participants under 65 were well-nourished, while 55% were undernourished. Conversely, among those aged 65 and above, 63% were well-nourished, and 37% were undernourished.. Malnutrition of female patients was higher than male in the cohort (60% vs 42%, p<0.001). Malnourished patients had a higher number of co morbidities compared to well-nourished patients (60.2% vs. 51.7%, p = 0.04). According to co morbidity category, 39.8% of malnourished patients had no co morbidity, whereas 60.2% had at least one co morbidity.

From preexisting medical conditions diabetic mellitus (31.32%) is the most common cause of co morbidity then hypertension (25.6%), other diseases (17%), lung diseases (9.8%), heart diseases (3.09%), HIV (1.7%), and cancer (0.2%) were listed from most to list commonest. Almost three-fourth (72.4%) of malnourished patients on admission were those who took multiple drugs (>5 number of drugs). Greater number patients who took poly medication was seen among malnourished when compared with well nourished (72.3% vs 59.6%, p<0.001)

## Nutritional status and clinical outcome among COVID-19 patients

Table 2 shows the nutritional status with their clinical outcome among COVID-19 patients. Larger percentage of ICU admission were seen among malnourished compared with well-nourished (24.2% vs 3.4%, p>0.001). The death analysis result showed that significant difference were seen between malnourished and well-nourished (5.2% vs 0.3%, p<0.001). In addition, Fig 1 shows underweight patients had a larger percentage of ICU admission rate than over weight patients (29.4% vs22.1%, p<0.001).Also the death of underweight patients were higher than over weight patients (7.1% vs 4.4%, p<0.001).

**Table 1. General admission-related characteristics of the patients according to nutrition status at baseline among COVID-19 patients in SPHMMC, EKGH and MCCC, Addis Ababa, 2021.**

| Variables | Well nourished | | Malnourished | | P-value |
|---|---|---|---|---|---|
| (N = 581) | Frequency | Percent | Frequency | Percent | |
| Age Group | | | | | <0.001* |
| <65 | 182 | 62.3% | 225 | 77.9% | |
| > = 65 | 110 | 37.7% | 64 | 22.1% | |
| Sex | | | | | <0.001* |
| Male | 199 | 68.2% | 147 | 50.9% | |
| Female | 93 | 31.8% | 142 | 49.1% | |
| Residence | | | | | 0.270 |
| Urban | 261 | 89.4% | 266 | 92.0% | |
| Rural | 31 | 10.6% | 23 | 8.0% | |
| Marital Status | | | | | 0.604 |
| Single | 41 | 14.0% | 45 | 15.6% | |
| Married | 251 | 86.0% | 244 | 84.4% | |
| Living arrangement | | | | | 0.986 |
| Alone | 48 | 16.4% | 49 | 17.0% | |
| With a partner | 8 | 2.7% | 8 | 2.8% | |
| With a parent/s | 236 | 80.8% | 232 | 80.3% | |
| Income | | | | | <0.001* |
| 0–600 | 51 | 17.5% | 49 | 17.0% | |
| 601–1650 | 4 | 1.4% | 22 | 7.6% | |
| 1651–3200 | 34 | 11.6% | 70 | 24.2% | |
| 3201–5250 | 82 | 28.1% | 79 | 27.3% | |
| 5251–7800 | 61 | 20.9% | 26 | 9.0% | |
| 7801–10900 | 50 | 17.1% | 29 | 10.0% | |
| >10,900 | 10 | 3.4% | 14 | 4.8% | |
| Alcohol drinking status | | | | | 0.336 |
| Yes | 65 | 22.3% | 55 | 19% | |
| No | 227 | 77.7% | 234 | 81% | |
| Smoking status | | | | | 0.600 |
| Yes | 36 | 12.3% | 46 | 15.9% | |
| No | 256 | 87.7% | 243 | 84.1% | |
| Comorbidity | | | | | 0.039* |
| No co-morbidity | 141 | 48.3% | 115 | 39.8% | |
| At least one co morbidity | 151 | 51.7% | 174 | 60.2% | |
| List of co-morbidities | N = 292 | | N = 289 | | |
| Diabetic mellitus | 84 | 28.8% | 101 | 34.9% | |
| Hypertension | 59 | 20.2% | 90 | 31.1% | |
| Other | 44 | 15.1% | 55 | 19.0% | |
| Lung diseases | 37 | 12.7% | 20 | 6.9% | |
| Heart diseases | 5 | 1.7% | 13 | 4.5% | |
| HIV AIDS | 0 | 0% | 5 | 1.7% | |
| Cancer | 0 | 0% | 1 | 0.3% | |
| Number of Medication | | | | | <0.001 |
| < = 5 | 118 | 40.4% | 80 | 27.7% | |
| >5 | 174 | 59.6% | 209 | 72.3% | |

Abbreviations; N: number (frequency): %: percent

## Effect of malnutrition on ICU admission and death

During the study period, a total of 16 patient fatalities were recorded, each of whom had been admitted to the Intensive Care Unit (ICU) prior to succumbing. Within this cohort, 15 deaths were observed in the malnourished group, while only one occurred in the well-nourished cohort. Upon further stratification of the malnourished subgroup, the analysis disclosed that 9 individuals were classified as overweight, while 6 were identified as undernourished, all of whom experienced mortality. Table 3 shows multivariable cox regression analysis model fitted to evaluate the effect of malnutrition on ICU admission and death. The Cox proportional hazard model was used for the analysis of nutritional status and COVID-19 adverse outcomes (ICU admission and death). Cox regression Adjusted Hazard Ratio for ICU Admission considering well-nourished as a reference. Show that the rate of ICU admission was significantly associated and were found to be higher among underweight (AHR = 10.02, 95% CI : (8.64–12.10) and overweight (AHR = 7.7, 95% CI : (6.41–9.62) patients when compared with well-nourished ones. But the age, sex, co morbidities and medication variables in the analysis were not significantly associated.

Table 4 shows Cox proportional hazard analysis on predictor of death for COVID patients. The rate of survival probability was significantly associated and were found to be better among well-nourished patients (AHR = 0.06, 95% CI : (0.01–0.44) when compared with malnourished COVID-19 patients.

Fig 2 shows Kaplan-Meir curve comparing ICU admission between nutritional statuses of COVID-19 patients. ICU admission rate of COVID-19 patients was plotted using Kaplan-Meier survival estimates. It shows that there was a statistically significant difference in survival time occurrence of ICU admission between underweight, overweight and well-nourished patients (log rank test P<0.001).

Fig 3 shows Kaplan-Meir curve comparing death between nutritional statuses of COVID-19.The Kaplan Meier survival estimates plotted for the death rate of COVID-19 patients. The result show that there was a statistically significant difference in survival time occurrence of death between malnourished and well-nourished patients (log rank test P<0.001).

## Discussion

The aim of the study was to assess the effect of malnutrition on ICU admission and death of COVID-19 patients in SPHMMC, EKGH, and MCCC. As a result, this study suggests that malnutrition increases the risk of ICU admission and death of COVID-19 patients. The study showed that there was a higher ICU admission and death proportion among the malnourished compared with the well-nourished. When compared to well-nourished patients, the rate of ICU admission was significantly associated and found to be higher among underweight [(AHR = 10.02, 95% CI: (8.64–12.10)] and overweight [(AHR = 7.7, 95% CI: (6.41–9.62)] patients. The rate of death was significantly associated with and was found to be lower among well-nourished patients [(AHR = 0.06, 95% CI : (0.01–0.44)] when compared with malnourished patients.

**Table 2. Nutritional status with their clinical outcome among COVID-19 patients in SPHMMC, EKGH and MCCC, Addis Ababa, 2021.**

| Out Come (N = 517) | Well nourished | | Malnourished | | P-value |
| --- | --- | --- | --- | --- | --- |
| | | | | | < 0.001* |
| | Frequency | Percent | Frequency | Percent | |
| ICU | 10 | 3.4% | 70 | 24.2% | |
| Death | 1 | 0.3% | 15 | 5.2% | |

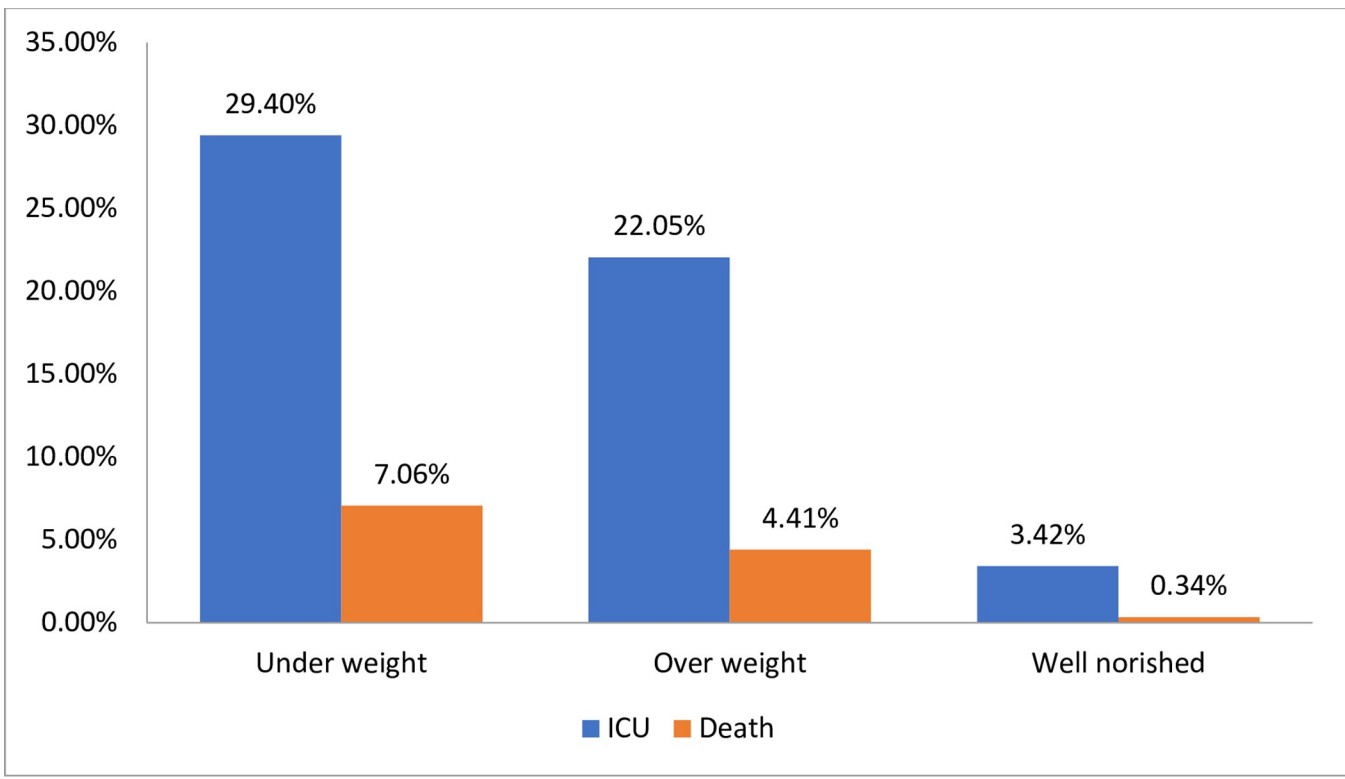

**Fig 1. Nutritional status with their clinical outcome among COVID-19 patients in SPHMMC, EKGH and MCCC, Addis Ababa, 2021.**

Similar previous studies show malnutrition is often doesn't gate enough attention due to being under-reported and often poorly documented. Malnutrition in developing countries including Ethiopia tends to refer to children and awareness about adult disease-related malnutrition is lack, so it stills a major problem in a clinical setting. Because of lack of knowledge, routine nutrition screening, assessment and intervention practices are not uniformly done as a part of medical care [23]. This study has tried to address this gap by showing the magnitude of the association between malnutrition and COVID-19 adverse outcomes. In order to recommend mandatory nutritional screening at admission for COVID-19 patients, it is necessary to give appropriate nutritional support.

The analysis of this study showed that being underweight has resulted in a higher rate of ICU admission. The result is consistent with other studies conducted on the effect of underweight on ICU admission. The result of an observational study done in France showed that the prevalence of malnutrition reached 66.7% in patients admitted to ICU [7]. A study done in Italy by Loris et al. in 2020 showed that malnutrition was present in 77% of those admitted to ICU [24]. A retrospective cohort analysis was done in China by enrolling 523 patients. Out of these, 211 individuals (40.3%) admitted to the ICU have a lower BMI [25]. Numerous study conducted in different countries suggests that a higher ICU admission rate was recorded among underweight patients. Due to the lack of malnutrition screening and assessment protocols for COVID-19 patients at admission and inadequate knowledge of health professionals specifically to diagnose malnutrition while at admission.

From the cox regression analysis results, the deaths of well-nourished COVID-19 patients were lower than malnourished patients. A retrospective and single-center study done in China showed that patients with moderate-to-severe malnutrition died at a higher rate than patients

**Table 3. Results of Cox proportional hazard analysis on predictor of ICU admission for patients with corona virus disease admitted at SPHMMC, EKGH and MCCC, Addis Ababa, 2021.**

| Variables | Category | Crude HR (95% CI) | Adjusted HR* (95% CI) |
|---|---|---|---|
| Comorbidity | | | |
| | No co-morbidity | 1.22 (0.78,1.9) | 1.11(0.70,1.76) |
| | At least one co-morbidity | 1 | 1 |
| Number of medications | | | |
| | ≤5 medication | 1.17(0.73,1.87) | 0.94(0.58,1.52) |
| | >5 medication | 1 | 1 |
| Age Group | | | |
| | <65 years | 0.62(0.32,1.06) | 0.79(0.46,1.37) |
| | ≥ 65 years | 1 | 1 |
| Sex | | | |
| | Male | 1.08(0.69,1.69) | 0.75(0.47,1.19) |
| | Female | 1 | 1 |
| Nutritional status | | | |
| | Well nourished | **1** | **1** |
| | Under weight | **9.96(8.14,11.76)** | **10.02(8.64,12.10)**** |
| | Over weight | **7.29(6.17,9.49)** | **7.66(6.41,9.62)**** |

** P-value < 0.05

**Table 4. Results of Cox proportional hazard analysis on predictor of death for patients with corona virus disease admitted at SPHMMC, EKGH and MCCC, Addis Ababa, 2021.**

| Variables | Category | Crude HR (95% CI) | Adjusted HR* (95% CI) |
|---|---|---|---|
| Co-morbidity | | | |
| | No co-morbidity | 1 | 1 |
| | At least one co-morbidity | 0.62(0.23,1.69) | 0.77(0.28,2.15) |
| Number of medications | | | |
| | ≤5 medication | 1 | 1 |
| | >5 medication | 0.60(0.19,1.88) | 0.86(0.27,2.72) |
| Age Group | | | |
| | <65 years | 1 | 1 |
| | ≥ 65 years | 0.96(0.33,2.76) | 0.72(0.25,2.10) |
| Sex | | | |
| | Male | 0.65(0.24,1.72) | 1.09(0.40,2.95) |
| | Female | 1 | 1 |
| Nutritional status | | | |
| | Well nourished | **0.06(0.01,0.44)** | **0.06(0.01,0.44)**** |
| | Malnourished | **1** | **1** |

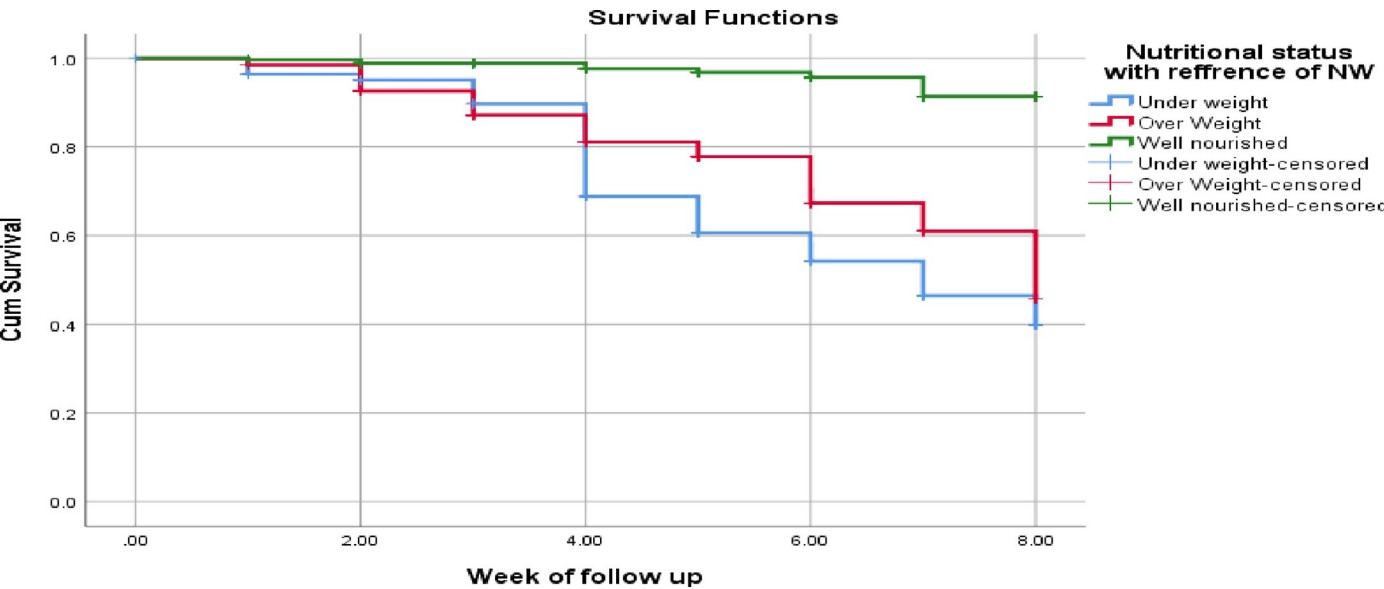

**Fig 2. Kaplan-Meir curve comparing ICU admission between nutritional statuses of COVID-19 patients (p < 0.01) admitted at SPHMMC, EKGH and MCCC, Addis Ababa, 2021.**

with normal or mild malnutrition [26]. The results of two studies conducted in Italy and France showed that well-nourished COVID-19 patients have better survival opportunities than malnourished ones [27, 28]. Therefore, evaluating the nutritional status of COVID_19 patients precisely is helpful for health professionals to identify the disease progression and

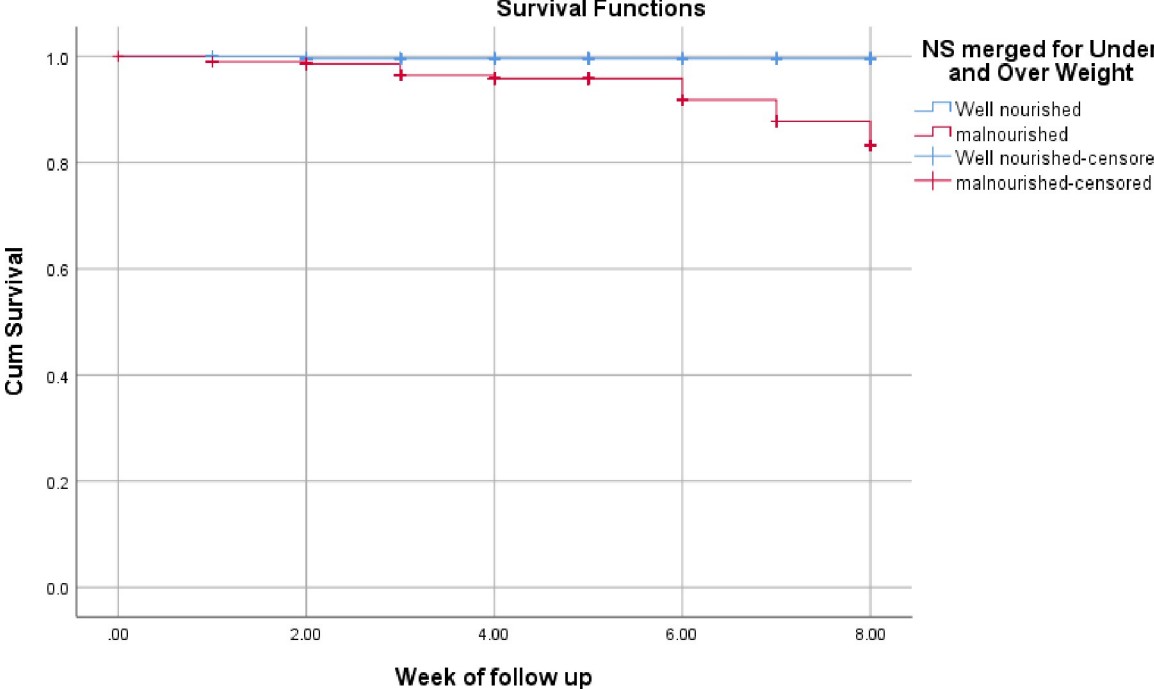

**Fig 3. Kaplan-Meir curve comparing death between nutritional statuses of COVID-19 patients (p < 0.01) admitted at SPHMMC, EKGH and MCCC, Addis Ababa, 2021.**

apply an appropriate therapy strategy in order to reduce the risk of ICU admission and death among malnourished patients.

The nutritional status of the COVID-19 patients at admission is an independent risk factor and has a significant effect on ICU admission and death of COVID-19 patients, which is supported by the literature [7, 10–13, 24, 29]. The study's findings indicate that being overweight during admission was highly related to ICU admission and death, even when other possible confounders were controlled for, and this finding was consistent with other studies. For example, a study done in Italy showed a higher need for assisted ventilation beyond pure oxygen support and a higher admission to intensive or semi-intensive care units were observed in patients with overweight and obesity even after adjusting for sex, age, and co-morbidities [10].

A retrospective cohort study conducted in the USA showed that severely obese COVID-19 patients (BMI $\geq$ 35 kg/m2) were 5.39 times more likely to be admitted to ICU when compared to patients with normal weight [12]. Another study done in China on the impact of obesity on disease severity and fatality among patients with COVID-19 showed that obesity raises the risk of hospitalization, ICU admission, invasive mechanical ventilation (IMV) requirements, and death [29].

A retrospective cohort study done in New York, USA showed that patients with overweight and obesity who have COVID-19 are at increased risk for mortality and intubation compared to those with normal BMI [11]. Similarly, a systematic review conducted in the United Kingdom to investigate if patients with obesity are more likely to die from COVID-19 compared to non-obese individuals showed that obesity is a risk factor for the death of COVID-19 patients [13]. Aside from nutritional status, factors such as age, gender, co-morbidities, and medication were not found to be statistically significant in the study's analysis, but previous research has found an association with worsening COVID-19 outcomes [3, 30–32].

In this study, the data analyzed were genuine anthropometric measurements, or primary data. This study addresses the hidden burden and consequence of malnutrition at admission in COVID-19 patients. As a result, it can be used to persuade hospitals and other healthcare facilities to adopt a strategy and a set of guidelines for identifying patients who may have nutritional risks and the study emphasizes the importance of early nutritional screening and to promote sufficient attention on nutritional intervention through developing guidance for covid-19 nutritional management.

Several limitations should be acknowledged in interpreting this research. The hospital designated for treating COVID-19 patients was government-assigned, and admissions were referrals from various government health facilities, potentially introducing selection bias. Furthermore, as the sole isolation and treatment center in the city with well-equipped intensive care units, there is a risk of bias in patient recruitment. Left truncation or delayed entry may contribute to selection bias as only patients who had not experienced the event were selectively entered at the time of study recruitment. Censoring, akin to loss of follow-up, can impact the study results by yielding incomplete data on participants. Despite efforts to minimize measurement error through rigorous training of data collectors and regular calibration of measurement tools, concerns about measurement accuracy persist.

## Conclusions

Malnutrition at the time of admission was shown to increase the risk of ICU admission and mortality among COVID-19 patients. Also, well-nourished COVID-19 patients have better survival opportunities than malnourished patients. To reduce the effects on patients and the healthcare system, it is crucial to assess patients' nutritional status as soon as they are admitted and to implement the proper nutritional therapy.

## Supporting information

**S1 File. Data set.**
(XLS)

**S1 Table. Socio-demographic characteristics of COVID-19 patients in SPHMMC, EKGH and MCCC, Addis Ababa, 2021.**
(DOCX)

## Acknowledgments

We would like to thank Addis Ababa University for facilitating the conduct of the research. We would also like to thank the study participants, the supervisors and data collectors without whom this wouldn't have been realized.

## Author Contributions

**Conceptualization:** Lencho Mekonnen Jima, Gudina Egeta Atomsa, Yakob Desalegn Nigatu.

**Data curation:** Lencho Mekonnen Jima, Gudina Egeta Atomsa, Yakob Desalegn Nigatu.

**Formal analysis:** Lencho Mekonnen Jima, Gudina Egeta Atomsa, Yakob Desalegn Nigatu.

**Investigation:** Lencho Mekonnen Jima, Gudina Egeta Atomsa, Yakob Desalegn Nigatu.

**Methodology:** Lencho Mekonnen Jima, Gudina Egeta Atomsa, Yakob Desalegn Nigatu.

**Resources:** Lencho Mekonnen Jima, Gudina Egeta Atomsa, Yakob Desalegn Nigatu.

**Supervision:** Lencho Mekonnen Jima.

**Visualization:** Lencho Mekonnen Jima, Gudina Egeta Atomsa, Yakob Desalegn Nigatu.

**Writing – original draft:** Yakob Desalegn Nigatu.

**Writing – review & editing:** Gudina Egeta Atomsa, Johane P. Allard, Yakob Desalegn Nigatu.

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
