## [Decision Letter · Decision Letter 0]

12 Dec 2022

PONE-D-22-22988The Effect of Malnutrition on Adult Covid-19 Patient’s ICU Admission and Mortality in Covid-19 Isolation and Treatment Centers in Ethiopia : A Prospective Cohort StudyPLOS ONE

Dear Dr. Desalegn,

Thank you for submitting your manuscript to PLOS ONE. After careful consideration, we feel that it has merit but does not fully meet PLOS ONE’s publication criteria as it currently stands. Therefore, we invite you to submit a revised version of the manuscript that addresses the points raised during the review process.

We look forward to receiving your revised manuscript.

Kind regards,

Wondwossen Amogne Degu, M.D

Academic Editor

PLOS ONE

Journal Requirements:

"NO - Include this sentence at the end of your statement: The funders had no role in study design, data collection and analysis, decision to publish, or preparation of the manuscript."

Additional Editor Comments (if provided):

Dear Yakob Desalegn

Greetings

The paper was reviewed by two independent reviewers and the final decision is that it needs major revision.

Attached below you find the reviewers comments.

(Reviewer 1)

Reviewer Recommendation Term: Major Revision

Rate Review: 0

Custom Review Question(s): Response

Comments to the Author

1. Is the manuscript technically sound, and do the data support the conclusions?

The manuscript must describe a technically sound piece of scientific research with data that supports the conclusions. Experiments must have been conducted rigorously, with appropriate controls, replication, and sample sizes. The conclusions must be drawn appropriately based on the data presented. Partly

2. Has the statistical analysis been performed appropriately and rigorously? No

3. Have the authors made all data underlying the findings in their manuscript fully available?

The PLOS Data policy requires authors to make all data underlying the findings described in their manuscript fully available without restriction, with rare exception (please refer to the Data Availability Statement in the manuscript PDF file). The data should be provided as part of the manuscript or its supporting information, or deposited to a public repository. For example, in addition to summary statistics, the data points behind means, medians and variance measures should be available. If there are restrictions on publicly sharing data—e.g. participant privacy or use of data from a third party—those must be specified. No

4. Is the manuscript presented in an intelligible fashion and written in standard English?

PLOS ONE does not copyedit accepted manuscripts, so the language in submitted articles must be clear, correct, and unambiguous. Any typographical or grammatical errors should be corrected at revision, so please note any specific errors here. Yes

5. Review Comments to the Author

(Reviewer 2)

Reviewer Recommendation Term: Major Revision

Rate Review: 0

Custom Review Question(s): Response

Comments to the Author

1. Is the manuscript technically sound, and do the data support the conclusions?

The manuscript must describe a technically sound piece of scientific research with data that supports the conclusions. Experiments must have been conducted rigorously, with appropriate controls, replication, and sample sizes. The conclusions must be drawn appropriately based on the data presented. No

2. Has the statistical analysis been performed appropriately and rigorously? No

3. Have the authors made all data underlying the findings in their manuscript fully available?

The PLOS Data policy requires authors to make all data underlying the findings described in their manuscript fully available without restriction, with rare exception (please refer to the Data Availability Statement in the manuscript PDF file). The data should be provided as part of the manuscript or its supporting information, or deposited to a public repository. For example, in addition to summary statistics, the data points behind means, medians and variance measures should be available. If there are restrictions on publicly sharing data—e.g. participant privacy or use of data from a third party—those must be specified. No

4. Is the manuscript presented in an intelligible fashion and written in standard English?

PLOS ONE does not copyedit accepted manuscripts, so the language in submitted articles must be clear, correct, and unambiguous. Any typographical or grammatical errors should be corrected at revision, so please note any specific errors here. No

5. Review Comments to the Author

Please use the space provided to explain your answers to the questions above. You may also include additional comments for the author, including concerns about dual publication, research ethics, or publication ethics. (Please upload your review as an attachment if it exceeds 20,000 characters) This paper is based on information gathered in Addis Ababa during the COVID-19 epidemic. It tries to find out if malnourished people are more likely to be admitted and more likely to die.

The School of Public Health and the COVID-19 treatment centres gave ethical approval for this study. I don't think these institutions have the power to give ethical approval for a study like this in Ethiopia. Based on what I read on the Addis Ababa University website on December 4th, 2022, I think it should have been the College of Health Sciences and not one of its departments.

The authors say that they were willing to give the data for the study, and they also say that all the data are described in the results. That is not true. For example, we don't know how BMI and MUAC are distributed. I think this paper shouldn't be published until the authors can give full access to all the data they've collected.

The main mistake the authors make is that they call malnutrition both under- and over-nutrition. Also, there is no way to explain how malnutrition was mdefined. It looks like they used both MUAC and BMI to figure if they were malnourshed. But these measures can't be switched around, and the authors don't talk about that.

A well-known risk factor for COVID admission and death is being overweight. But undernutrition is a group of different problems with many causes. It is important for the authors to explain why the patients were undernourished and what medical problems caused them to be thin. The fact that they used more medicines than the other patients suggests that there were underlying causes for their undernutrition.

Does this paper add any new knowledge? Unless the authors can provide much more details about the undernourished group, I do not think so.

6. PLOS authors have the option to publish the peer review history of their article (what does this mean?). If published, this will include your full peer review and any attached files.

Do you want your identity to be public for this peer review? For information about this choice, including consent withdrawal, please see our Privacy Policy.

Yes: Bernt Lindtjorn

Confidential to Editor

1. Do you have any potential or perceived competing interests that may influence your review? Please review our Competing Interests policy and declare any potential interests that you feel the Editor should be aware of when considering your review. If you have no competing interests, please write "I have no competing interests." No

2. Did you receive any assistance in preparing this review (e.g. from a post-doc or graduate student)? If yes, please include their name below.

3. If accepted, do you think this submission should be highlighted on the PLOS ONE website? PLOS ONE does not evaluate manuscripts based on perceived significance or readership. We aim to provide tools for readers to filter and evaluate our publications. (optional) No

Do you want to get recognition for this review on a Web of Science researcher profile?

If you opt in, your Web of Science profile will automatically be updated to show a verified record of this review in full compliance with the journal’s review policy. If you don’t have a Web of Science profile, you will be prompted to create a free account. Yes

More Reviewer Details

The authors present relevant and unique data on associations between various forms of malnutrition and severe outcomes (ICU and/or death) among patients presenting to a single Ethiopia referral hospital with a Covid-19 diagnosis.

Several major concerns need to be addressed.

1. Malnutrition is a composite category, and the authors need to be clear and consistent. In this paper, both undernutrition and overweight were assessed, but not other kinds of malnutrition (eg micronutrient deficiencies). This is not a problem, but clear definitions need to be given by BMI and/or MAUC, and both types of malnutrition must not be lumped anywhere, when describing associations or assessing risks. These are different populations.

2. BMI>25 and in particular BMI>30 are already well-established risk factors for adverse outcomes following Covid-19 infection. Thus, the added value of this paper is more in showing, or contrasting the risks associated with undernutrition. This would give a more clear perspective to introduction and discussion, and the paper overall.

3. The authors need to present a more thoughtful discussion on implications of their findings, beyond providing unclear statements on 'suitable interventions to be provided'. It is not clear what interventions they want to see implemented, when, why, nor to what extent there are evidence-based interventions, in particular upon hospitalisation, which improve outcomes assessed. Are they suggestion to put obese patients on a strict diet&exercise plan? Force-feeding undernourished patients?

4. Also, it is remarkable that some risk factors observed in other settings (co-morbidities, age) were not significantly associated with adverse outcomes here. This can cast doubt on the validity of the data collected and conclusions reached, and needs at least some thoughtful reflection.

5. In contrast to many other underlying conditions, manifest undernutrition and overnutrition are not very difficult to assess visually by trained staff, thus the statement that these condition are often unrecognised is unclear. Perhaps the authors meant that the associations with of increased risk for adverse outcomes are not always appreciated, but either way it would be good to have references to back up such unexpected claims.

6. More information needs to be given on the Covid-19 policies in the setting. Were these all patients referred, or were they a mix of direct presentations and referrals? Were all patients included because of their Covid-19 related clinical state, or were there also patients included who presented with an unrelated problem, but were routinely tested and found infected?

7. The presentation of data needs to be much improved. Baseline data on BMI and MUAC need to be presented by outcome category. Clarification needs to be given to what extent deaths occurred at ICU or elsewhere, and how data on patients who were first admitted to ICU, and later died (either at ICU or in the ward were analysed. Were there cases where patients opted to go home to die? While the KM curves provide some key information on time to events, it does not clarify patients with both events, and these findings are not properly mentioned in the text. Are there background data on the prevalence of obesity and undernutrition in the catchment populations?

8. The authors also need to reflect on potential non-causal explanations for the association between undernutrition and adverse outcomes. Eg, is it possible that undernourished people have less easy access to health services (eg due to poverty, rural residence, civil unrest, with more time to reach/less available health services, etc), and might they therefore be presenting at a more severe or later stadium, resulting in worse outcomes? Was there information available on distance of the residence to the hospital, and on duration of complaints prior to arrival? How many critically ill (defined as? Assessed by someone indepent from the study team?) were excluded?

9. Please clarify why MUAC was used in addition/as alternative to BMI, as inclusion was restricted to adults, and critically ill patients for whom nutritional status could not be assessed at admission were excluded already?

10. A major overhaul is needed to reduce/eliminate the frequent repetitions and superfluous sentences where no relevant information is presented; also in part of the methods there is an overload of details relating to standard procedures, which is unnecessary; and in parts of the results there is quite a bit of repetition of non-essential data already presented in the tables/figures, rather than highlighting the main findings only in the text.

Some minor comments:

The Abstract mentions in the results that the mean age was 55 years, (16,45) years. Please clarify the 16,45 (also in the Results section).

Methods: it seems a bit odd to justify the measurements referring to a (non-peer reviewed) publication related from the Australian sports federation (ref 14). Is that a relevant context to provide validity to an Ethiopian hospital setting.

Methods: with a mean age of 55, it is surprising that 81% were were categorised as living with parents, and less than 3% with a partner. Presumably, elderly parents might be living with their grown up children, with or without partner? However, in general it is not clear what the added value of this category is, suggest to drop.

Results: unclear why ages were split in only 2 categories, please do at least a sensitivity analysis with eg 4 categories. On the other hand, 7 categories for income do not seem useful, suggest to reduce to eg 4. For smoking and alcohol: clarify if this refers to current situation or ever. The high number of people taking over 5 medications (60%) appears at odds with 48% reporting no co-morbidity, can you explain? More information on what co-morbidities might be helpful.

6. PLOS authors have the option to publish the peer review history of their article (what does this mean?). If published, this will include your full peer review and any attached files.

Do you want your identity to be public for this peer review? For information about this choice, including consent withdrawal, please see our Privacy Policy.

No

Confidential to Editor

1. Do you have any potential or perceived competing interests that may influence your review? Please review our Competing Interests policy and declare any potential interests that you feel the Editor should be aware of when considering your review. If you have no competing interests, please write "I have no competing interests." I have no competing interests.

2. Did you receive any assistance in preparing this review (e.g. from a post-doc or graduate student)? If yes, please include their name below. No

3. If accepted, do you think this submission should be highlighted on the PLOS ONE website? PLOS ONE does not evaluate manuscripts based on perceived significance or readership. We aim to provide tools for readers to filter and evaluate our publications. (optional) Yes, on a more specific subject area page (e.g. Biochemistry, Atmospheric Science)

Do you want to get recognition for this review on a Web of Science researcher profile?

If you opt in, your Web of Science profile will automatically be updated to show a verified record of this review in full compliance with the journal’s review policy. If you don’t have a Web of Science profile, you will be prompted to create a free account. No

Wondwossen Amogne MD, PhD

Academic editor, PLOS ONE

Reviewers' comments:

Reviewer's Responses to Questions

**Comments to the Author**

1. Is the manuscript technically sound, and do the data support the conclusions?

Reviewer #1: Partly

Reviewer #2: No

2. Has the statistical analysis been performed appropriately and rigorously? 

Reviewer #1: No

Reviewer #2: No

3. Have the authors made all data underlying the findings in their manuscript fully available?

Reviewer #1: No

Reviewer #2: No

4. Is the manuscript presented in an intelligible fashion and written in standard English?

Reviewer #1: Yes

Reviewer #2: No

5. Review Comments to the Author

Reviewer #1: The authors present relevant and unique data on associations between various forms of malnutrition and severe outcomes (ICU and/or death) among patients presenting to a single Ethiopia referral hospital with a Covid-19 diagnosis.

Several major concerns need to be addressed.

1. Malnutrition is a composite category, and the authors need to be clear and consistent. In this paper, both undernutrition and overweight were assessed, but not other kinds of malnutrition (eg micronutrient deficiencies). This is not a problem, but clear definitions need to be given by BMI and/or MAUC, and both types of malnutrition must not be lumped anywhere, when describing associations or assessing risks. These are different populations.

2. BMI>25 and in particular BMI>30 are already well-established risk factors for adverse outcomes following Covid-19 infection. Thus, the added value of this paper is more in showing, or contrasting the risks associated with undernutrition. This would give a more clear perspective to introduction and discussion, and the paper overall.

3. The authors need to present a more thoughtful discussion on implications of their findings, beyond providing unclear statements on 'suitable interventions to be provided'. It is not clear what interventions they want to see implemented, when, why, nor to what extent there are evidence-based interventions, in particular upon hospitalisation, which improve outcomes assessed. Are they suggestion to put obese patients on a strict diet&exercise plan? Force-feeding undernourished patients?

4. Also, it is remarkable that some risk factors observed in other settings (co-morbidities, age) were not significantly associated with adverse outcomes here. This can cast doubt on the validity of the data collected and conclusions reached, and needs at least some thoughtful reflection.

5. In contrast to many other underlying conditions, manifest undernutrition and overnutrition are not very difficult to assess visually by trained staff, thus the statement that these condition are often unrecognised is unclear. Perhaps the authors meant that the associations with of increased risk for adverse outcomes are not always appreciated, but either way it would be good to have references to back up such unexpected claims.

6. More information needs to be given on the Covid-19 policies in the setting. Were these all patients referred, or were they a mix of direct presentations and referrals? Were all patients included because of their Covid-19 related clinical state, or were there also patients included who presented with an unrelated problem, but were routinely tested and found infected?

7. The presentation of data needs to be much improved. Baseline data on BMI and MUAC need to be presented by outcome category. Clarification needs to be given to what extent deaths occurred at ICU or elsewhere, and how data on patients who were first admitted to ICU, and later died (either at ICU or in the ward were analysed. Were there cases where patients opted to go home to die? While the KM curves provide some key information on time to events, it does not clarify patients with both events, and these findings are not properly mentioned in the text. Are there background data on the prevalence of obesity and undernutrition in the catchment populations?

8. The authors also need to reflect on potential non-causal explanations for the association between undernutrition and adverse outcomes. Eg, is it possible that undernourished people have less easy access to health services (eg due to poverty, rural residence, civil unrest, with more time to reach/less available health services, etc), and might they therefore be presenting at a more severe or later stadium, resulting in worse outcomes? Was there information available on distance of the residence to the hospital, and on duration of complaints prior to arrival? How many critically ill (defined as? Assessed by someone indepent from the study team?) were excluded?

9. Please clarify why MUAC was used in addition/as alternative to BMI, as inclusion was restricted to adults, and critically ill patients for whom nutritional status could not be assessed at admission were excluded already?

10. A major overhaul is needed to reduce/eliminate the frequent repetitions and superfluous sentences where no relevant information is presented; also in part of the methods there is an overload of details relating to standard procedures, which is unnecessary; and in parts of the results there is quite a bit of repetition of non-essential data already presented in the tables/figures, rather than highlighting the main findings only in the text.

Some minor comments:

The Abstract mentions in the results that the mean age was 55 years, (16,45) years. Please clarify the 16,45 (also in the Results section).

Methods: it seems a bit odd to justify the measurements referring to a (non-peer reviewed) publication related from the Australian sports federation (ref 14). Is that a relevant context to provide validity to an Ethiopian hospital setting.

Methods: with a mean age of 55, it is surprising that 81% were were categorised as living with parents, and less than 3% with a partner. Presumably, elderly parents might be living with their grown up children, with or without partner? However, in general it is not clear what the added value of this category is, suggest to drop.

Results: unclear why ages were split in only 2 categories, please do at least a sensitivity analysis with eg 4 categories. On the other hand, 7 categories for income do not seem useful, suggest to reduce to eg 4. For smoking and alcohol: clarify if this refers to current situation or ever. The high number of people taking over 5 medications (60%) appears at odds with 48% reporting no co-morbidity, can you explain? More information on what co-morbidities might be helpful.

Reviewer #2: This paper is based on information gathered in Addis Ababa during the COVID-19 epidemic. It tries to find out if malnourished people are more likely to be admitted and more likely to die.

The School of Public Health and the COVID-19 treatment centres gave ethical approval for this study. I don't think these institutions have the power to give ethical approval for a study like this in Ethiopia. Based on what I read on the Addis Ababa University website on December 4th, 2022, I think it should have been the College of Health Sciences and not one of its departments.

The authors say that they were willing to give the data for the study, and they also say that all the data are described in the results. That is not true. For example, we don't know how BMI and MUAC are distributed. I think this paper shouldn't be published until the authors can give full access to all the data they've collected.

The main mistake the authors make is that they call malnutrition both under- and over-nutrition. Also, there is no way to explain how malnutrition was mdefined. It looks like they used both MUAC and BMI to figure if they were malnourshed. But these measures can't be switched around, and the authors don't talk about that.

A well-known risk factor for COVID admission and death is being overweight. But undernutrition is a group of different problems with many causes. It is important for the authors to explain why the patients were undernourished and what medical problems caused them to be thin. The fact that they used more medicines than the other patients suggests that there were underlying causes for their undernutrition.

Does this paper add any new knowledge? Unless the authors can provide much more details about the undernourished group, I do not think so.

6. PLOS authors have the option to publish the peer review history of their article (what does this mean?). If published, this will include your full peer review and any attached files.

Reviewer #1: No

Reviewer #2: **Yes: **Bernt Lindtjorn

---

## [Author Response · Author response to Decision Letter 0]

24 Feb 2023

A rebuttal letter 

Dear PLOS ONE's Journal Editor

We would like to express our gratitude for giving us another opportunity to revise the manuscript as per reviewer comments. We affirm that the manuscript has improved a lot from your comments. 

We hereby listed Point-by-point responses within the 'Response to Reviewer'.

Kind regards,

'Response to Reviewers

Editor Comments:

Address: Dear reviewer, Thank you for the comment. We have corrected it as recommended. We have checked carefully to ensure the PLOS ONE’S requirements.

State what role the funders took in the study. If the funders had no role in your study, please state: “The funders had no role in study design, data collection and analysis, decision to publish, or preparation of the manuscript.”

Address: Dear reviewer, Thank you for the comment. We have corrected it as recommended (line number 333)

Additional Editor Comments:

Reviewer comments

Reviewer #1: The authors present relevant and unique data on associations between various forms of malnutrition and severe outcomes (ICU and/or death) among patients presenting to a single Ethiopia referral hospital with a Covid-19 diagnosis.

Several major concerns need to be addressed.

1. Malnutrition is a composite category, and the authors need to be clear and consistent. In this paper, both under nutrition and overweight were assessed, but not other kinds of malnutrition (eg micronutrient deficiencies). This is not a problem, but clear definitions need to be given by BMI and/or MAUC, and both types of malnutrition must not be lumped anywhere, when describing associations or assessing risks. These are different populations.

Address: Dear reviewer, Thank you for the comment. There was clear operational definition for malnutrition using BMI & MUACwith the help of ref [15].Undernourished: COVID-19 case with BMI <18.5 kg/m2and for those critically ill patients who are not capable of measuring BMI the MUAC measurement of<23.7 were considered as underweight (15). Overnourished:COVID-19 case with BMI >25 kg/m2and for those critically ill patients who are not capable of measuring BMI the MUAC measurement of ≥28.1were considered as overweight (15).

2. BMI>25 and in particular BMI>30 are already well-established risk factors for adverse outcomes following Covid-19 infection. Thus, the added value of this paper is more in showing, or contrasting the risks associated with undernutrition. This would give a more clear perspective to introduction and discussion, and the paper overall.

Address: Dear reviewer, Thank you for the comment. The add value of this research paper is more on showing the adverse effect of undernourished on COVID-19 patients. As you suggested it is elaborated in detail in discussion section of the paper. 

3. The authors need to present a more thoughtful discussion on implications of their findings, beyond providing unclear statements on 'suitable interventions to be provided'. It is not clear what interventions they want to see implemented, when, why, nor to what extent there are evidence-based interventions, in particular upon hospitalization, which improve outcomes assessed. Are they suggestion to put obese patients on a strict diet&exercise plan? Force-feeding undernourished patients?

Address: Dear reviewer, Thank you so much for your valuable comments.The study emphasizes the importance of early nutritional screening and to promote sufficient attention on nutritional intervention through developing guidance for covid-19 nutritional management.

4. Also, it is remarkable that some risk factors observed in other settings (co-morbidities, age) were not significantly associated with adverse outcomes here. This can cast doubt on the validity of the data collected and conclusions reached, and needs at least some thoughtful reflection.

Address: Dear reviewer, Thank you for your insightful comment, our research haven’t revealed any association between co-morbidity and adverse COVID-19 this might be due to the sample size was calculated based on the proportion of adverse effect of COVID-19 patient among the malnourished and well-nourished patients.

5. In contrast to many other underlying conditions, manifest undernutrition and overnutrition are not very difficult to assess visually by trained staff, thus the statement that these condition are often unrecognised is unclear. Perhaps the authors meant that the associations with of increased risk for adverse outcomes are not always appreciated, but either way it would be good to have references to back up such unexpected claims.

Address: Thank you again for your valuable comments. Saying left unrecognized is an ambiguous phrase so we have to change into “doesn’t gate enough attention”this can explain it well. Besides there is a literature which suggests Malnutrition in developing countries including Ethiopia tends to refer to children and awareness about adult disease-related malnutrition is lack, so it stills a major problem in a clinical setting. Because of lack of knowledge, routine nutrition screening, assessment and intervention practices are not uniformly done as a part of medical care (21)

6. More information needs to be given on the Covid-19 policies in the setting. Were these all patients referred, or were they a mix of direct presentations and referrals? Were all patients included because of their Covid-19 related clinical state, or were there also patients included who presented with an unrelated problem, but were routinely tested and found infected?

Address: Dear reviewer, Thank you so much for your valuable comments. The centers selected for the study was designated for covid-19 patients treatment only. So the other health institution in the city can referred covid-19 cases to these centers for isolation, well-equipped ICU setup and for their larger admission capacity.But some referred caseswere found while routine covid-19 tests.

7. The presentation of data needs to be much improved. Baseline data on BMI and MUAC need to be presented by outcome category. Clarification needs to be given to what extent deaths occurred at ICU or elsewhere, and how data on patients who were first admitted to ICU, and later died (either at ICU or in the ward were analyzed. Were there cases where patients opted to go home to die? While the KM curves provide some key information on time to events, it does not clarify patients with both events, and these findings are not properly mentioned in the text. Are there background data on the prevalence of obesity and undernutrition in the catchment populations?

Address: Dear reviewer, Thank you so much for your valuable comments.There is no case consideration of ICU admission after death outcome .For further clarification those patients admitted to ICU, and later died while follow up were reported as death outcome.ICU admission and death was censored at 2 months, hence COVID-19 patients who were discharged earlier than follow up time without adverse outcome or stayed in the hospital for more than 2 months were treated as censored observations. There was no any background data on the prevalence of obesity and undernutrition in the catchment population.

8. The authors also need to reflect on potential non-causal explanations for the association between undernutrition and adverse outcomes. Eg, is it possible that undernourished people have less easy access to health services (eg due to poverty, rural residence, civil unrest, with more time to reach/less available health services, etc), and might they therefore be presenting at a more severe or later stadium, resulting in worse outcomes? Was there information available on distance of the residence to the hospital, and on duration of complaints prior to arrival? How many critically ill (defined as? Assessed by someone indepent from the study team?) Were excluded?

Address: Dear reviewer, Thank you so much for your valuable comments. It is bit difficult to reflect some of the factor which may have effect on the association of under nutrition and adverse outcome because of during base line data collection the mentioned factors weren’t included in the study. We didn’t include the onset of the disease because the patients were enrolled based the severity of the disease at admission.Individual who full fill the exclusion criteria they haven’t been registered by the data collectors. So there is no compiled number which shows the number of excluded patients who are critically ill at admission. 

9. Please clarify why MUAC was used in addition/as alternative to BMI, as inclusion was restricted to adults, and critically ill patients for whom nutritional status could not be assessed at admission were excluded already?

Address: Dear reviewer, Thank you so much for the comment. Wrongly wrote the sentence in line 88 as we all know MUAC measurement was need to measure critical ill patients who are not capable of measuring their BMI. So we put the correct sentence 'critically ill patients with both upper extremities amputation were excluded’.MUAC is also recommended for patients with ascites or edema in legs or trunk to gauge dry weight and BMI(16).Other studies confirm that MUAC correlated well with BMI and could be used to identify patients as underweight and overweight(17-20).

10. A major overhaul is needed to reduce/eliminate the frequent repetitions and superfluous sentences where no relevant information is presented; also in part of the methods there is an overload of details relating to standard procedures, which is unnecessary; and in parts of the results there is quite a bit of repetition of non-essential data already presented in the tables/figures, rather than highlighting the main findings only in the text.

Address: Dear reviewer, Thank you for the comments. We agree with you sir. So we have addressed those comments.

Some minor comments:

1. The Abstract mentions in the results that the mean age was 55 years, (16,45) years. Please clarify the 16,45 (also in the Results section).

Address: Dear reviewer, Thank you for the comments. The sentence in line 166 show that the mean age of study sample was 55 years and the standard deviation was 16.45 years.

2. Methods: it seems a bit odd to justify the measurements referring to a (non-peer reviewed) publication related from the Australian sports federation (ref 14). Is that a relevant context to provide validity to an Ethiopian hospital setting.

Address: Dear reviewer, There is no study which shows acceptable TEM level in Ethiopian hospital setting.

3. Methods: with a mean age of 55, it is surprising that 81% were were categorised as living with parents, and less than 3% with a partner. Presumably, elderly parents might be living with their grown up children, with or without partner? However, in general it is not clear what the added value of this category is, suggest to drop.

Address: Dear reviewer, Thank you for the comments. Yes it was confusing, so we have to omit the sentence in line 173-175 general.

4. Results: unclear why ages were split in only 2 categories, please do at least a sensitivity analysis with eg 4 categories. On the other hand, 7 categories for income do not seem useful, suggest to reduce to eg 4.

Address: Dear reviewer, Thank you for the comments. From earlier research adults over 65 years of age represent 80% of hospitalizations and have a 23-fold greater risk of death than those under 65.this is the main reason for age categorization.

5. For smoking and alcohol: clarify if this refers to current situation or ever. The high number of people taking over 5 medications (60%) appears at odds with 48% reporting no co-morbidity, can you explain? More information on what co-morbidities might be helpful.

Address: Dear reviewer, Thank you for the comments. The substance status refers to ever smoking and alcohol drinking status. Even if larger number of covid-19 patients doesn’t have co-morbidity covid-19 disease by itself need supportive medications/supplements like vitamin A, C, zinc and iron as a result they take over 5 medications.

---

## [Decision Letter · Decision Letter 1]

8 Nov 2023

PONE-D-22-22988R1The Effect of Malnutrition on Adult Covid-19 Patient’s ICU Admission and Mortality in Covid-19 Isolation and Treatment Centers in Ethiopia : A Prospective Cohort StudyPLOS ONE

Dear Dr. Desalegn,

Thank you for submitting your manuscript to PLOS ONE. After careful consideration, we feel that it has merit but does not fully meet PLOS ONE’s publication criteria as it currently stands. Therefore, we invite you to submit a revised version of the manuscript that addresses the points raised during the review process.

We look forward to receiving your revised manuscript.

Kind regards,

Nitai Roy

Guest Editor

PLOS ONE

Reviewers' comments:

Reviewer's Responses to Questions

**Comments to the Author**

1. If the authors have adequately addressed your comments raised in a previous round of review and you feel that this manuscript is now acceptable for publication, you may indicate that here to bypass the “Comments to the Author” section, enter your conflict of interest statement in the “Confidential to Editor” section, and submit your "Accept" recommendation.

Reviewer #3: (No Response)

Reviewer #4: (No Response)

2. Is the manuscript technically sound, and do the data support the conclusions?

Reviewer #3: Partly

Reviewer #4: Yes

3. Has the statistical analysis been performed appropriately and rigorously? 

Reviewer #3: No

Reviewer #4: No

4. Have the authors made all data underlying the findings in their manuscript fully available?

Reviewer #3: No

Reviewer #4: No

5. Is the manuscript presented in an intelligible fashion and written in standard English?

Reviewer #3: Yes

Reviewer #4: No

6. Review Comments to the Author

Reviewer #3: The authors have conducted a prospective study evaluating the impact of malnutrition on COVID-19 outcomes. The study is important as not many studies have assessed malnutrition in COVID-19, especially in the African continent. Unfortunately the manuscript in its current form has issues with the presentation and interpretation of statistics which need to be addressed before it can be further assessed.

Introduction

1. line 54: Correct "SARS-Cov2 infection (COVID-19) and related pneumonia" to "SARS-CoV-2 infection and related pneumonia (COVID-19)"

Materials and Methods

2. line 85: "All COVID-19 cases with confirmed positive COVID-19 test result were included..." What type of SARS-CoV-2 tests were used, eg. RT-PCR or antigen? The line should read SARS-CoV-2 test rather than COVID-19 test. How were patients with SARS-CoV-2 infection differentiated from those who specifically had COVID-19 pneumonia?

3. line 86: "Patients were excluded from the study if they are critically with both upper extremities amputation ill..." Correct this sentence for meaning.

4. line 99: "Based on this assumption, a total of 581 COVID-19 patients were included in the study based on the highest sample size obtained from ICU admission prevalence." As a prospective study, it would be better to state that 581 COVID-19 patients were required rather than included.

5. line 128: "MUAC measurement of<23.7 were considered" Provide unit of measurement for MUAC each time it is used.

Results

6. line 177: "There were statistically significant association seen between age, sex, income, co morbidity,

medication and nutritional status." This sentence could be more specific, eg. "There were statistically significant association seen between nutritional status and younger age, female sex, income, co morbidity, and higher number of medication."

7. line 178: "More percentage of patients < 65 years age group was seen in both malnourished (77.9%) and well-nourished 182 (62.3%) nutritional status." This statement is non-contributory as it only means more patients aged < 65 we recruited compared to > 65. It would be more helpful to give the percentage of patients under and over 65 who were well nourished and malnourished (ie. use the number of patients in each age group as the denominator) to enable comparisons.

8. line 179: "Malnutrition of male patients was higher than female in the cohort (50.9% vs 49.1%, p<0.001)." It is true that the absolute number of male patients who were malnourished is higher, but the proportion of female patients who are malnourished is actually higher (42% vs 60%). The authors have again used the incorrect denominator. The same is true for the other comparisons made in this paragraph.

9. line 184: "From preexisting medical conditions diabetic mellitus (31.32%) is the most common cause of co

morbidity then hypertension (25.6%), other diseases (17%), lung diseases (9.8%)..." Consider including specific comorbidities in table 1, divided by nutritional status. Several of these comorbidities are known to be risk factors for poor outcomes in COVID-19 and their effects on the findings of this study need to be considered.

10. line 195: "Nutritional status and clinical outcome among COVID-19 Patients". It is difficult to assess the accuracy of the statistics provided here as no absolute numbers are provided. It is also not stated how many patients were over weight. Over weight patients seem to be separated in some places, and grouped together with well-nourished patients in others, such as the preceding paragraphs and table 1. The authors need to be more consistent with how patients are grouped.

Discussion

11. line 263: "Of these, 211(40.3%) were admitted to the ICU as a result of lower BMI(23)." More accurate to state that they had lower BMI, not that the admission was as a result of the lower BMI.

12. line 264: "Every study conducted in different". Numerous studies, but not every study. Many studies did not report underweight patients.

Other

13. Figure 1: The percentages only add up to 66.68%. What happened to the remainder?

Reviewer #4: This manuscript reports the findings of a multi-center observational study of assessing the effect of malnutrition on Covid-19 outcome at Covid centers in Ethiopia. The manuscript is generally well written, and data of this type remain very unusual for Africa despite major concerns were detected on the methods and analysis. I hope the authors will find the following comments helpful.

The introduction needs some minor language editing.

Study design: The authors dictated they have conducted a prospective cohort despite this unusually they have 0% drop out. Furthermore, the authors follow the patients until 2 months since the follow-up seems long there was no withdrawal which is impractical.

Since the study was prospective cohort all 581 patents were followed for 2 months but was there a patient died being admitted to ICU? The authors should perform a sub-group analysis for malnutrition and death based on ICU admission status.

How many patients were admitted to ICU and died during the follow-up? How the authors treat those patients defer admission to ICU but decided to be admitted by the attending physician?

Why patients were followed for 2 months? Was there any justification?

The authors should cite paper for the auumption used for double population formula.

Statistical analysis: A survival analysis model was used to determine whether there was association between malnutrition and adverse Covid-19 outcome. However proper statistical procedures were not employed. The dependent variable should be time to event (event= admission to ICU or death). I strongly Did the authors check the proportional hazard assumption (using Schonefeld residual or log-log plot) before fitting the regression model? If so, it should have to be reported. The authors also chel the dataset they used fit for the model they fit (use Co-Snell or Martingale residuals). Was the Log-rank test significant for other variables which were adjusted in the multivariable model since if the variable is not significant on the Log-rank test it is not necessary to include in the Cox regression model. A sensitivity analysis should be performed sine the events were few.

Results: the results were a little confusing the authors should present 1. The baseline characteristics 2. The exposure and outcome tabulation 3. The regression model results. The figures showing the Kaplan-Meier survival curve are not visible. The result of Cox-regression model ton assesses the effect of malnutrition on Covid-19 mortality shows there was no difference between the crude and adjusted hazard ratios this indicate the variable (Malnourished/ Well-nourished) was not confounded by other variables this indicated the baseline characteristics were similar which is unlikely and results to be very questionable.

Discussion: This may be clearer to the reader if it adhered more closely to a standard four paragraph structure: Para 1: summaries main findings, Para 2: compare and contrast with previous work, Para 3: strengths and limitations, Para 4: conclusions and future work. The authors cover most of this already but I think the limitations could be more detailed.

Thanks for the opportunity to review this paper.

7. PLOS authors have the option to publish the peer review history of their article (what does this mean?). If published, this will include your full peer review and any attached files.

Reviewer #3: No

Reviewer #4: No

---

## [Author Response · Author response to Decision Letter 1]

5 Jan 2024

A rebuttal letter 

Dear PLOS ONE's Journal Editor

We would like to express our gratitude for giving us another opportunity to revise the manuscript as per reviewer comments. We affirm that the manuscript has improved a lot from your comments. 

We hereby listed Point-by-point responses within the 'Response to Reviewer'.

Kind regards,

*N.B, the line number we mentioned here is based on the revised track change manuscript file

'Response to Reviewers

Reviewer comments

Reviewer #3:

1. Line 54: Correct "SARS-Cov2 infection (COVID-19) and related pneumonia" to "SARS-CoV-2 infection and related pneumonia (COVID-19)"

Address: Dear reviewer, Thank you for your valuable suggestion. We have duly revised the statement in accordance with your guidance. (Page 3, line number 54).

2. Line 85: "All COVID-19 cases with confirmed positive COVID-19 test result were included..." What type of SARS-CoV-2 tests were used, eg. RT-PCR or antigen? The line should read SARS-CoV-2 test rather than COVID-19 test. How were patients with SARS-CoV-2 infection differentiated from those who specifically had COVID-19 pneumonia?

Address: Thank you for bringing this matter to our attention in the main research paper we have addressed the issue by explicitly stating that we utilized the rRT-PCR test for diagnosing the SARS-CoV-2 virus. This clarification has also been incorporated into the track change document for your reference. (Page 5, line number 89-91).

3. Line 86: "Patients were excluded from the study if they are critically with both upper extremities amputation ill..." Correct this sentence for meaning.

Address: Dear reviewer, Thank you for your valuable suggestion. We have revised the sentences to enhance clarity and meaning, aligning with your feedback (page 5, line number 92-94)

4. Line 99: "Based on this assumption, a total of 581 COVID-19 patients were included in the study based on the highest sample size obtained from ICU admission prevalence." As a prospective study, it would be better to state that 581 COVID-19 patients were required rather than included.

Address: Thank you for your valuable suggestion. In response, we have refined the statement to convey a more accurate representation. Instead of stating that 581 COVID-19 patients were included, we now specify that this number was required for the study based on the highest sample size obtained from ICU admission prevalence (page 5, line number 105)

.

5. Line 128: "MUAC measurement of<23.7 were considered" Provide unit of measurement for MUAC each time it is used.

Address: Thank you for bringing this matter to our attention. We have addressed the issue, and the corrected measurement is now recorded as 23.7 cm. (page 6, line number 134)

6. Line 177: "There were statistically significant association seen between age, sex, income, co morbidity, medication and nutritional status." This sentence could be more specific, e.g. "There were statistically significant association seen between nutritional status and younger age, female sex, income, co morbidity, and higher number of medications."

Address: Dear reviewer, Thank you for your suggestion. We have refined the sentence to specify a statistically significant association between nutritional status and younger age, female sex, income, co-morbidity, and a higher number of medications (page 9, line number 191-192).

7. Line 178: "More percentage of patients < 65 years age group was seen in both malnourished (77.9%) and well-nourished 182 (62.3%) nutritional status." This statement is non-contributory as it only means more patients aged < 65 we recruited compared to > 65. It would be more helpful to give the percentage of patients under and over 65 who were well nourished and malnourished (i.e. use the number of patients in each age group as the denominator) to enable comparisons.

Address: Thank you for your constructive comment. In response, we have made the suggested revision and incorporated it with track changes for your review. (Page 9, line number 192-195).

8. Line 179: "Malnutrition of male patients was higher than female in the cohort (50.9% vs 49.1%, p<0.001)." It is true that the absolute number of male patients who were malnourished is higher, but the proportion of female patients who are malnourished is actually higher (42% vs 60%). The authors have again used the incorrect denominator. The same is true for the other comparisons made in this paragraph.

Address: Thank you for highlighting this issue. We have addressed it by making the recommended revision, and you can find the changes incorporated in the track changes for your review (Page 9, line number 196-197).

9. Line 184: "From preexisting medical conditions diabetic mellitus (31.32%) is the most common cause of co-morbidity then hypertension (25.6%), other diseases (17%), lung diseases (9.8%)..." Consider including specific comorbidities in table 1, divided by nutritional status. Several of these comorbidities are known to be risk factors for poor outcomes in COVID-19 and their effects on the findings of this study need to be considered.

Address: Dear reviewer, Thank you for your comment. We have addressed the feedback by incorporating the listed co-morbidities into Table 1 (Page 11-12).

10. Line 195: "Nutritional status and clinical outcome among COVID-19 Patients". It is difficult to assess the accuracy of the statistics provided here as no absolute numbers are provided. It is also not stated how many patients were overweight. Overweight patients seem to be separated in some places, and grouped together with well-nourished patients in others, such as the preceding paragraphs and table 1. The authors need to be more consistent with how patients are grouped.

Address: Thank you for your comment. The absolute numbers of both malnourished and well-nourished cases have been specified as recommended (Page 9, line number 189-190).

11. Line 263: "Of these, 211(40.3%) were admitted to the ICU as a result of lower BMI (23)." More accurate to state that they had lower BMI, not that the admission was as a result of the lower BMI.

Address: Dear reviewer, Thank you for your comment. We have rephrased the statement according your suggestion. (Page 18, line number 291-292).

12. Line 264: "Every study conducted in different". Numerous studies, but not every study. Many studies did not report underweight patients.

Address: Dear reviewer, your comment is absolutely right and we have corrected it as recommended (Page 18, line number 293-294).

13. Figure 1: The percentages only add up to 66.68%. What happened to the remainder?

Address: Dear reviewer, figure 1 only shows the percentage of ICU admission and death among malnourished and well-nourished patients, out of the total COVID-19 patients enrolled. The remaining were neither admitted to the ICU nor died. (The figure only shows patients with an outcome)

Reviewer #4

1. The introduction needs some minor language editing.

Address: Dear reviewer, we have revised the whole manuscript for spelling and grammar.

2. The authors dictated they have conducted a prospective cohort despite this unusually they have 0% drop out. Furthermore, the authors follow the patients until 2 months since the follow-up seems long there was no withdrawal which is impractical.

Address: Dear reviewer, your comment is absolutely right and we understand your concern. During the initial days of the COVID-19 pandemic in Ethiopia, a limited number of isolation and treatment centers were available, wherein patients underwent isolation for several weeks. The isolation duration extended until two consecutive rRT-PCR tests returned negative results, at which point patients were considered cleared. This regulatory requirement compelled patients to remain in isolation until the manifestation of the desired outcome. Consequently, there was no challenge in maintaining follow-up within our cohort.

3. Since the study was prospective cohort all 581 patents were followed for 2 months but was there a patient died being admitted to ICU? The authors should perform a sub-group analysis for malnutrition and death based on ICU admission status.

Address: Thank you for your insightful comment, we did not include cases where ICU admission occurred after the patient's demise in our analysis of death outcomes. For additional clarification, instances involving patients admitted to the ICU who subsequently passed away during the follow-up period were classified and reported as death outcomes.

4. How many patients were admitted to ICU and died during the follow-up? How the authors treat those patients defer admission to ICU but decided to be admitted by the attending physician? Why patients were followed for 2 months? Was there any justification?

Address: Dear reviewer, Thank you for identifying this flow. In our efforts to rectify the issue, we have revised the statement to provide a clearer depiction of the overall mortality and subcategories based on nutritional status. The total number of deaths recorded during the study period was 16, comprising 9 individuals classified as overweight, 6 identified as undernourished, and 1 classified as well-nourished. Notably, all patients who succumbed had been admitted to the Intensive Care Unit (ICU) prior to their demise. 

This period allows for monitoring the protracted recovery period and identifying persistent symptoms like fatigue and neurological issues. Additionally, it enables the assessment of severe outcomes such death and ICU admission. On top of this period allows us to identify long-term complications, including those emerging weeks after recovery.

5. The authors should cite paper for the assumption used for double population formula.

Address: Dear reviewer, the assumptions regarding sample size were initially included in the main document, but for manuscript purposes, they were omitted. However, in the current version, these assumptions have been reintroduced and appropriately cited. The basis for determining the prevalence of ICU admission and death among COVID-19 patients was derived from articles 31 and 32 (Page 24, line number 454-458).

6. Statistical analysis: A survival analysis model was used to determine whether there was association between malnutrition and adverse Covid-19 outcome. However proper statistical procedures were not employed. The dependent variable should be time to event (event= admission to ICU or death). I strongly Did the authors check the proportional hazard assumption (using Schonefeld residual or log-log plot) before fitting the regression model? If so, it should have to be reported.

Address: Dear reviewer, Thank you for your insightful comment. In our analysis conducted using SPSS software, both Schoenfeld residuals and log-log plots were employed to assess the proportional hazard assumption, specifically examining the parallelism between survival curves for the variables of interest. This comprehensive evaluation aimed to ensure the constancy of hazard ratios over time. The results of these diagnostic checks are duly reported in our study as recommended (Page 8, line number 159-161).

7. The results were a little confusing the authors should present 1. The baseline characteristics 2. The exposure and outcome tabulation 3. The regression model results. The figures showing the Kaplan-Meier survival curve are not visible.

Address: Thank you for your valuable feedback. While the baseline characteristics and regression details were initially lacking elaboration in the manuscript, I want to highlight that the exposure and outcome tabulations, which were previously omitted, have now been incorporated in the latest version. This addition enhances the comprehensiveness of the manuscript and addresses the noted omission.

8. The result of Cox-regression model ton assesses the effect of malnutrition on Covid-19 mortality shows there was no difference between the crude and adjusted hazard ratios this indicate the variable (Malnourished/ Well-nourished) was not confounded by other variables this indicated the baseline characteristics were similar which is unlikely and results to be very questionable.

Address: Dear reviewer, Thank you for your valuable comment. The observed similarity in the crude and adjusted hazard ratios concerning death reports among well-nourished and malnourished patients can be attributed to the limited number of recorded deaths during the study period. Specifically, the occurrence of only one death within the well-nourished cohort may have led to statistical challenges and influenced the stability of hazard ratio estimates.

9. This may be clearer to the reader if it adhered more closely to a standard four paragraph structure: Para 1: summaries main findings, Para 2: compare and contrast with previous work, Para 3: strengths and limitations, Para 4: conclusions and future work. The authors cover most of this already but I think the limitations could be more detailed.

Address: Dear reviewer, Thank you for your valuable input. In response to your guidance, we have made efforts to concisely address the limitations within the discussion section, adhering to the goal of brevity as suggested.

---

## [Decision Letter · Decision Letter 2]

22 Jan 2024

The Effect of Malnutrition on Adult Covid-19 Patient’s ICU Admission and Mortality in Covid-19 Isolation and Treatment Centers in Ethiopia : A Prospective Cohort Study

PONE-D-22-22988R2

Dear Dr.  Desalegn,

We’re pleased to inform you that your manuscript has been judged scientifically suitable for publication and will be formally accepted for publication once it meets all outstanding technical requirements.

Kind regards,

Nitai Roy

Guest Editor

PLOS ONE

Additional Editor Comments (optional):

Reviewers' comments:

Reviewer's Responses to Questions

**Comments to the Author**

1. If the authors have adequately addressed your comments raised in a previous round of review and you feel that this manuscript is now acceptable for publication, you may indicate that here to bypass the “Comments to the Author” section, enter your conflict of interest statement in the “Confidential to Editor” section, and submit your "Accept" recommendation.

Reviewer #4: All comments have been addressed

Reviewer #5: (No Response)

2. Is the manuscript technically sound, and do the data support the conclusions?

Reviewer #4: Yes

Reviewer #5: Yes

3. Has the statistical analysis been performed appropriately and rigorously? 

Reviewer #4: Yes

Reviewer #5: Yes

4. Have the authors made all data underlying the findings in their manuscript fully available?

Reviewer #4: Yes

Reviewer #5: Yes

5. Is the manuscript presented in an intelligible fashion and written in standard English?

Reviewer #4: Yes

Reviewer #5: Yes

6. Review Comments to the Author

Reviewer #4: Authors tried their best to amend the manuscript based on the comments provided, I would like to appreciate that. The only minor concern I have is that the authors must cite a literature where did they get the assumption for sample size calculation “……P1(Proportion of ICU admission among malnourished 96 patients) =9.5%, P2(Proportion of ICU admission among well-nourished patients) =9.5%.also P1 97 and P2 assumed for death P1 (Proportion of death among malnourished patients)=1.5%,P2 98 (Proportion of death among well-nourished patients)=1.5% ,Z α/2 =1.96 at a 95% level of 99 confidence, 5% margin of error…….”. Merit the publication of this manuscript given several limitations outlined since studies in critical care outcome is rare in the low-resource setting like Ethiopia I don’t have further questions and I would like to congratulate the authors for doing this work.

Reviewer #5: (No Response)

7. PLOS authors have the option to publish the peer review history of their article (what does this mean?). If published, this will include your full peer review and any attached files.

Reviewer #4: **Yes: **Amanuel Sisay Endeshaw

Reviewer #5: **Yes: **Md Ekhtear Hossain

---

## [Editor Report · Acceptance letter]

8 Mar 2024

PONE-D-22-22988R2 

PLOS ONE

Dear Dr. Nigatu, 

I'm pleased to inform you that your manuscript has been deemed suitable for publication in PLOS ONE. Congratulations! Your manuscript is now being handed over to our production team.

Kind regards, 

on behalf of

Dr. Nitai Roy 

Guest Editor

PLOS ONE